# Systematic Review of Tumor Segmentation Strategies for Bone Metastases

**DOI:** 10.3390/cancers15061750

**Published:** 2023-03-14

**Authors:** Iromi R. Paranavithana, David Stirling, Montserrat Ros, Matthew Field

**Affiliations:** 1Faculty of Engineering and Information Sciences, School of Electrical, Computer and Telecommunications Engineering, University of Wollongong, Wollongong, NSW 2522, Australia; 2Ingham Institute for Applied Medical Research, Liverpool, NSW 2170, Australia; 3Southwestern Sydney Cancer Services, NSW Health, Sydney, NSW 2170, Australia; 4School of Clinical Medicine, Southwestern Sydney Clinical Campus, UNSW, Sydney, NSW 2170, Australia

**Keywords:** bone metastases, computational approaches, deep learning, machine learning, malignant lesions, radiation therapy

## Abstract

**Simple Summary:**

With recent progress in radiation therapy, patients with bone metastases can be treated curatively, provided precise delineation of metastatic lesions is adequately identified. Tumor segmentation is a highly active area of research, but only limited studies have been on bone metastasis. This review aims to investigate methods for differentiating benign from malignant bone lesions and characterizing malignant bone lesions specifically in the context of bone metastases. While computer vision techniques have opened new opportunities for quantifying cancer growth with minimal expert supervision, fully automatic segmentation algorithms still require improvement. This is partly due to limited contrast between tumors and surrounding tissue and the lack of a widely agreed upon “gold standard” for defining these boundaries. Additionally, many studies do not provide evidence that their proposed methods are suitable for use in clinical practice.

**Abstract:**

Purpose: To investigate the segmentation approaches for bone metastases in differentiating benign from malignant bone lesions and characterizing malignant bone lesions. Method: The literature search was conducted in Scopus, PubMed, IEEE and MedLine, and Web of Science electronic databases following the guidelines of Preferred Reporting Items for Systematic Reviews and Meta-Analyses (PRISMA). A total of 77 original articles, 24 review articles, and 1 comparison paper published between January 2010 and March 2022 were included in the review. Results: The results showed that most studies used neural network-based approaches (58.44%) and CT-based imaging (50.65%) out of 77 original articles. However, the review highlights the lack of a gold standard for tumor boundaries and the need for manual correction of the segmentation output, which largely explains the absence of clinical translation studies. Moreover, only 19 studies (24.67%) specifically mentioned the feasibility of their proposed methods for use in clinical practice. Conclusion: Development of tumor segmentation techniques that combine anatomical information and metabolic activities is encouraging despite not having an optimal tumor segmentation method for all applications or can compensate for all the difficulties built into data limitations.

## 1. Introduction

Bone is one of the most common metastatic sites for cancer, especially in the lung, breast, and prostate [1]. This type of metastasis is often painful, with a high risk of mortality. The median survival rate of patients suffering from bone lesions metastasized from breast, prostate, and renal cancer ranges between 12 and 33 months, while survival is critically low for patients with primary lung cancer along with bone metastasis, ranging from 9.5% to 12% with one-year survival [1]. The level of bone metastasis is strongly linked with shorter survival rates [2]. Generally, these patients are treated with palliative chemotherapy and radiotherapy in clinical practice [3]. More recently, advances in image-guided radiotherapy techniques, such as stereotactic body radiotherapy (SBRT), have enabled the delivery of potentially ablative radiation doses while respecting healthy tissue constraints [4,5,6]. Furthermore, clinical trials, such as the SABR-COMET trial, have shown the benefits of SBRT for metastatic disease [7]. Effective treatment methods can improve overall survival and long-term progression-free survival [8]. Predictions of treatment response and feasibility can be improved by quantifying the number of metastatic lesions, their location, and the impact of radiomic biomarkers [9].

In medical image analysis, various modalities, such as positron emission tomography (PET) [10,11], whole-body Magnetic Resonance Imaging (MRI) [12], and bone scintigraphy [13], are used to support diagnoses and clinical follow-up. PET imaging offers functional details and is commonly used to evaluate cancer [14]. There are many advantages to using computed tomography (CT) in hybrid nuclear medicine equipment, such as attenuation correction and visual correlation between functional and anatomical images. Recent literature has shown that segmentation based on both CT and PET can determine the volume of interest (VOI) based on the anatomical contour [15]. Segmentation involves identifying the sets of pixels or voxels that form the tissue of interest [16]. Several reviews have been published reporting medical image segmentation methods, along with the strengths and weaknesses and discussing the challenges and outcomes [14,17,18,19,20,21,22,23,24,25,26,27,28]. A literature review by Sahiner et al. noted that establishing clinical significance is as important as establishing statistical significance for the research. Incorporating expert medical knowledge to optimize methods can provide benefits beyond adding extra layers to a Convolutional Neural Network (CNN) model and help radiologists accept the use of models [29]. Similarly, Zhang and Sejdić mentioned that although there are many applications for machine learning to help radiologists, they still cannot substitute for the clinician’s role due to existing limitations. One limitation is that many studies in radiology are based on supervised learning, and algorithms learn specific patterns based on radiologists’ decisions. Very few radiologists made these decisions in segmentation and were subject to varying degrees of inter-observer variability. Therefore, further investigations are needed to decide whether a machine can perform alone with 100% accuracy or at least match inter-observer variability [28]. In a review of deep learning segmentation for radiotherapy, Samarasinghe et al. noticed that clinical sites mostly used the U-net architecture and mostly the CT datasets from in-house data sources [30]. Further research contributions are needed to justify the use of algorithms in clinical decision-making to improve patient outcomes and translate them into clinical practice. It is difficult to interpret how models make decisions between input and output due to the significant number of parameters used. Acceptance of a model is improbable if medical experts cannot validate the approach and understand the logical bases of the method [19,28].

Faiella et al. [31] investigated the potential role of radiomics as a decision-supporting tool for predicting bone disease status, distinguishing benign from malignant bone lesions, and characterizing malignant lesions at the genetic level, considering only CT and MRI imaging. Their study is the first that we found while reviewing articles on bone metastases in the radiomics aspect. However, the paper lacks a discussion on segmentation techniques used in the radiomics approach. To date, no review has presented bone metastasis segmentation approaches and tumor segmentation with PET, CT, or PET/CT in radiation therapy to our knowledge. The main objective of this review was to present an overview of the latest research on cancer segmentation and bone metastasis segmentation on radiology images in the context of radiation therapy planning and to analyze and compare with state-of-the-art techniques in computer vision.

## 2. Methods

### 2.1. Literature Search

The systematic review followed the guidelines of the Preferred Reporting Items for Systematic Reviews and Meta-Analyses 2020 (PRISMA) [32]. The search was conducted in Scopus, PubMed, IEEE and MedLine, and the Web of Science electronic databases for publication date between 2010 and 2022. In addition, we used a Google search to identify additional records. The reference list of the included articles was cross-checked to identify any additional articles. All original studies published in English, with full text available, reporting bone metastasis segmentation or tumor segmentation for radiotherapy patients with oligometastatic disease, were included. The studies that contained segmentation as a part of their work were included. The studies that used CT, PET, and PET/CT were included. The studies that used bone scans could not be excluded because bone scans are primarily used to detect metastasis, as they appear as a hotspot on a bone scan. The studies, including medical implants, virtual clinical trials, image registration, MRI, and PET/MRI studies, were excluded. The awareness services for each issue of the journal were excluded. When several versions of the same articles were presented, the latest version was cited.

The databases were searched on 1 April 2022. The query was designed to include all studies that contained one or more words from four groups, one group comprised of words associated with bones (bone and bones), the second group comprised with the words associated with metastasis cancer (metastasis, metastases, metastatic, cancer, cancers, tumor, tumors, tumour, tumours, oligometastatic disease), the third group comprised with the words associated with radiotherapy (radiation oncology, radiation therapy, radiotherapy), and the fourth group with the term segmentation.

The complete search query used in the Scopus database was therefore:

“ALL (“bone and bones” OR “bone” OR “bones”) AND ALL (metastasis OR metastase-sor AND metastatic OR cancer OR cancers OR tumor OR tumors OR tumour OR tumours OR “oligometastatic disease”) AND ALL (“radiation oncology” OR “radiation therapy” OR radiotherapy) AND ALL (segmentation) AND PUBYEAR > 2009 AND PUBYEAR < 2023”.

An equivalent query with the same keywords was used in other databases. We used the following search query to identify the additional studies using Google search:

“bone metastasis segmentation”.

After excluding duplicate articles and assessing the remaining articles for eligibility based on their title and abstract, only relevant publications proceeded to full-text screening. The first author (I.R.P.) performed the screening, and the second author (M.F.) reviewed the screening.

### 2.2. Data Extraction

The outcomes of interest were the segmentation approaches used for cancerous tumors and bone metastasis segmentation. Data were extracted with regard to the following:Enrollment period of the patients;Study type: retrospective cohort study or prospective;Study population. Extracted the number of scans or images when patient numbers were not provided;Training/Validation/testing cohorts;Primary tumor and relevant location;Imaging modality;Methodology;Outcome;Evaluation Metrics;Details of whether the study mentioned the suitability of the approaches for clinical use;Country of the Authors.

## 3. Results

A flow diagram of the literature selection process is presented in Figure 1. We conducted a comprehensive literature search, using both databases and Google searches to identify relevant studies on bone metastasis segmentation. A total of 2513 articles were identified through the initial database search, with an additional 302 papers found through alternative sources not included in the initial search. After removing duplicates, 2524 records were screened based on their titles and abstracts. Of these, 2367 records were excluded due to inclusion/exclusion criteria, leaving 157 full-text articles for further inspection.

After a detailed assessment of the full-text articles, 55 were excluded due to incomplete information or not meeting the inclusion/exclusion criteria. The most common reasons for exclusion were articles related to image registration, medical implants, and MRI or PET/MR studies. Finally, we included 102 full-text articles in our systematic review, with 24 review articles and 1 comparison study article focusing on techniques and technologies used in medical imaging for image analysis and segmentation. These papers provided background information for our study, with the remaining 77 original studies being the focus of our analysis. The categorization of the included articles is summarized in Table 1.

Of these original studies, 18 included segmentation tasks on cancer metastasis, which accounted for 23.37%. We focused on the segmentation of metastasis and other areas separately to weigh the effort given to metastasis in past years. The number of studies that performed segmentation of OARs/organs, tumor, and Target Volumes/OARs, along with target volumes, were 22, 27, and 4, respectively, yielding 53 studies. The increasing pace of publications in tumor segmentation was observable in recent years.

Figure 2a shows that most of the papers used CT (50.65%, of 77 original works), while Figure 2b shows that 58.44% of papers included in our review were based on deep learning techniques, with the remaining papers using thresholding, classification, clustering, statistical, atlas-based, and region-based techniques. Figure 2c shows the number of papers over time, with a dramatic increase in the number of publications in 2020 and 2022.

We found that there was a lack of consistency in performance evaluation metrics, making cross-evaluation of segmentation approaches difficult (Figure 2d). As shown in Figure 2e, 27.27% (21 articles) of papers had first authors from China, while 16.88% (13 articles) had first authors from the United States. Collaborative research across multiple countries is crucial for advancing scientific knowledge and developing effective solutions to global challenges. However, our analysis of 77 original articles revealed that only a small proportion (18.18%) involved collaboration among authors from two [37,55,63,65,71,75,78,81,84,85,105] or three [70,91,92] different countries. This suggests that there is still a lack of international collaboration and data sharing in the field. It is important to encourage and facilitate such collaborations to foster the exchange of ideas, resources, and expertise, ultimately leading to more impactful research outcomes. Lung cancer was the most commonly used primary cancer type in the studies, with 21 articles, followed by prostate cancer at 11 (Figure 2f). However, 15 articles did not report the primary cancer type. This review presents different methods and approaches for the tumor segmentation problem, but not all of these methods have been rigorously tested in real-world clinical settings or validated against accepted standards or benchmarks. As a result, many proposed methods did not present sufficient evidence to demonstrate their suitability for widespread clinical use. Of 77 original articles, only 19 articles (24.67%) reported the feasibility of using their methods in a clinical setting yet required further study on the matter. The data were extracted from all original articles included in Appendix A.

## 4. Discussion

Several review papers in the present literature discussed deep learning, machine learning, and other techniques separately on PET, CT, PET/CT, or bone scintigraphy images [19,29]. Some focused only on an imaging type [14,18]. No individual article focused on a combination of different types of techniques applied to multiple imaging modalities. Existing reviews of segmentation approaches have focused on the concept of radiomics and identified some of the promising avenues for the future, both in terms of applications and technical innovations. However, these studies did not focus on whether the existing methods were feasible for use in clinical practice. The following sections discuss the broad approaches used for the segmentation.

### 4.1. Deep Learning

Most of the papers in the review used either CNN or U-Net (a modification of CNN) as the strategy for studies conducted with deep learning. CNNs were mostly used for applications such as identification, diagnosis, classification, and segmentation of bone metastasis [13,24,37,47,52,95,99,107,108], identification of critical regions associated with toxicities after liver SBRT [92], tumor co-segmentation [74,78], radiation dose calculations [112], and OAR segmentation [86,106].

Smaller datasets in deep learning can result in overfitting, and high-quality patient data are crucial to reduce bias in clinical practice. However, there are privacy and ethical concerns in handling medical data, and a lack of labeled data to train deep learning algorithms, making manual labeling expensive and requiring expertise from physicians. This task is also prone to uncertainty when physicians label multiple classes per lesion [56]. As a solution to this, Lin et al. [44] developed a single-photon emission computerized tomography (SPECT) image annotation system in their work based on the openly available tool LabelMe released by MIT (http://labelme.csail.mit.edu/Release3.0/) [113] for manual labeling of SPECT images, which has a low spatial resolution. Apiparakoon et al. [56] used a semi-supervised learning method, the Ladder Feature Pyramid Network (LFPN), which incorporates an autoencoder structure in the ladder network to self-train using the unlabeled data. Even though LFPN achieves a slightly lower F1-score alone than self-training, models with self-training require twice the training time than the semi-supervised approach. Some studies have also suggested pretraining the model with unlabeled data from related datasets to overcome the lack of labeled data [45,114].

Augmentation is another approach to addressing data limitations. Apiparakoon et al. [56] augmented a dataset by changing the light, contrast, and brightness to ensure consistency with the physician’s process. Several other augmentation techniques have been employed, including rescaling [13,33,39,43,44,45,54,58,59,60,98,107], rotation [13,33,39,43,44,45,54,58,59,60,77,98,107], zooming [13,44,107], shifting intensity [58,77], reflecting horizontally [77], translating the image [39,43,77], cropping [59], applying elastic deformations [45], gamma augmentation [45], and flipping [13,44,54,59,107]. Da Cruz et al. [60] further applied a probabilistic Gaussian blur and linear contrast filters to augment the dataset. Furthermore, Zhang et al. [77] used advanced augmentation methods: Mixup data augmentation, random erase operation, CutMix, and Mosaic method. Mixup [115] generates additional samples during the training process by convexly combining random pairs of images and their associated labels. The authors utilized Mixup to deal with significant memory loss and the network’s inadequate sensitivity to the symmetry of GANs. A random erase operation was performed on the data prior to the backbone network to prevent overfitting. All portions and locations erased are random for each round of training, with the erased section considered either a blocked or distorted portion processed by filling pixels with fixed color or filling with the mean of the RGB channel of all pixels. CutMix [116] was used to cut and paste the lesion areas to other background areas to improve learning of lesion features and to help learn positive features within unbalanced samples. The Mosaic [117] method employs several images simultaneously and can enrich the discovered objects’ backgrounds.

Transfer learning is another strategy that authors use to deal with limited data [77,84,108]. Most studies use datasets from the same source(s) for training and testing, so generalizability is not well studied. Feng et al. [90] discovered that a DCNN model trained on a public dataset performed poorly on the institution’s data due to differences in clinical practice. Retraining with local cases improved performance, with retraining from scratch being slightly more effective than transfer learning. No advantage was observed in collecting more training data for poor performance. In contrast, Protonotarios et al. [71] introduced a dynamic information fusion scheme by applying a few-shot learning (FSL) framework. The FSL approach built a user-centric model of re-training that constantly improves with end-user feedback. During deployment, the end user may assess the model’s outputs, and, if erroneous, they may correct them.

Han, Oh, and Lee [41] proposed two CNN architectures: (1) whole body-based (WB) and (2) tandem architectures using the whole-body bone scan and local 256 × 512 patches, followed by a final fully connected deep neural network for integrating global (i.e., whole-body) and local (i.e., patches) information, named “global-local unified emphasis” (GLUE), and both were trained on limited data. Compared with classical 2D CNN models, this model results in a higher performance on limited data in this case [41].

A limitation of the model-based approaches is potential biases in small datasets, such as studying gender-biased data [47]. Insufficient data to train the models was an issue in most of the studies in this review [13,39,44,45,47,65,90,105,106,107]. Lu et al., 2020 [66] adopted a strategy of transforming the image segmentation issue into a pixel-wise classification issue. Each pixel is regarded as an independent training sample during network training, increasing the sample size significantly. Furthermore, the authors promoted the high efficiency of network training and reduced overfitting simultaneously by applying Adam stochastic optimization [118] and batch normalization [119]. In various tasks, initializations of network weights are usually from models (such as VGG19) trained with the ImageNet dataset. Sartor et al. [106] indicated that more data and consistently annotated data were needed for their model to achieve higher CNN overlap and enable future clinical implementation. Additionally, Lou et al. [111] reported that their study could not account for all biases due to population heterogeneity in their datasets (due to clinical stage, radiation dose, CT scanners, and motion management) and the limited size of the independent validation cohort. Song et al. [50] found that noisy CT images caused false positive classifications of bone metastasis, and some areas of the lesion could not be detected. The authors suggest that building a 3D voxel detector may eliminate these issues.

Two papers in this review focused on automatic segmentation for treatment response [45] and treatment planning [89] for metastatic lesions. Moreau et al. [45] compared two methods for bone lesion segmentation in metastatic breast cancer based on the nn-Unet [120] architecture: (1) use of lesion annotations with PET and CT images as 2-channel input; (2) use of both the reference bone and lesion masks as ground truth. The use of bone masks improved precision and slightly improved the Dice score for bone lesion segmentation. Moreau et al. [45] also proposed two nn-Unet segmentation models to compute imaging biomarkers for treatment response from baseline and follow-up images. When manually segmenting or assessing treatment responses, experts usually look at both baseline and follow-up acquisitions to decide the patients’ responses. Therefore, two input channels, baseline PET images and lesion segmentation on the baseline PET, were added to the follow-up network. Four imaging biomarkers were computed from the manual and automatic segmentations, and these produced promising results for predicting the treatment response. Improved results can be obtained using multimodal imaging modalities like PET/CT [36]. Arends et al. [89] showed that automatic vertebral body delineation using CNN was of high quality, which can save time in a clinical radiotherapy workflow.

Deep neural networks are based on complex, inter-connected hierarchical representations of the training data; however, interpreting these representations is quite demanding [107]. While interpretability needs to be enhanced, the research community should further investigate how to measure sensitivity and visualize features. Model transparency and interpretability are important to explain the model, understand the value, and ensure the robustness of the findings. For instance, Apiparakoon et al. [56] stated that they extracted global features from the core network, but the features were not mentioned in the study. This makes it difficult to detect what the model focuses on and to provide explanations of why the model makes its categorizations. The generalizability of these methods also requires further evaluation to embed them in clinical decision support systems.

### 4.2. Thresholding

Thresholding is a simple segmentation technique that focuses on converting a gray-level image to a binary image by defining all the voxels greater (or lower) than a given value to be foreground and the remaining to be background [14]. Various types of thresholding methods, including fixed, iterative, adaptive, and regional, are used by authors for different applications, such as tumor segmentation, OAR segmentation, detection of increased uptake regions in bone scintigraphy, quantification of bone metastasis, and detection of bone lesions. Thresholding-based segmentation on PET/CT images is on the Hounsfield Unit (HU) and the Standard Uptake Value (SUV). A CT image voxel is in HU, which has a scale range between −1000 and approximately 30,000 [121]. DICOM stores the pixel values of images in 12- to 16-bit formats. CT threshold segmentation can target high-density regions, such as bone [122]. In contrast, image segmentation on PET scans based on thresholding employs intensity probabilities using image histograms. SUV is a normalized semiquantitative parameter that can be derived using the intensity of PET images and DICOM metadata, including acquisition time and dose of the radiotracer. The SUV is then used for image segmentation [14].

The thresholding-based papers reviewed in this article were used for detection, segmentation and quantification of bone, bone metastasis, and bone lesions [40,68,70,82,96,100]. Detection involves localizing organs, landmarks, or lesions in medical images [19,70,100] whereas segmentation is aimed at obtaining detailed boundaries of the structures [68,82,96,100]. Quantification of detected lesions or metastasis focuses on extracting features for further analysis. For example, total bone metastasis was quantified using total bone metastasis volume, percentage of affected bone tissue, SUVmean, and SUVmax in the affected tissue, Z-transformed deviation of SUV in the affected tissue from average SUV in nonaffected tissue, and total metastasis count [40].

Some authors used hybrid methods by combining thresholding with other methods, such as flood filling algorithms [82] and graph cut algorithms [96]. Fränzle et al. [82] built a fully automated shape model positioning for bone segmentation in whole-body CT scans using fixed thresholding for skeleton segmentation and a flood filling algorithm for segmentation of the medullary cavities inside the skeleton. The proposed method provides all the information needed for the automatic selection and initialization of a statistical shape model for long bone segmentation. Nguyen et al. [96] suggested a framework for segmenting spinal marrow compartments from full-body joint PET/CT scans acquired after bone marrow transplantation. It included three main components: full body graph cut segmentation, spinal column vertebral body segmentation, and cancellous region extraction.

The main limitations of studies that use thresholding for PET imaging involve low resolution with high contrast, the large variability of pathologies, inherent noise, and high uncertainties in fuzzy object boundaries. There is no consensus on the selection of an SUV threshold [14]. Tsujimoto et al. [100] showed that improvements can be made in setting the threshold values, especially by analyzing the feasibility of other thresholding techniques and threshold derivation algorithms. Hammes et al. [40] found that the HU threshold had no significant influence, whereas an SUV threshold of 2.5 proved optimal for automated lesion quantification. Lesions with intense tracer uptake might lead to errors in estimations of the total affected bone volume because that area might exceed the true anatomic borders of the lesion, causing overestimation of the affected bone volume. Moussallem et al. [68] identified that the main difficulty limiting the segmentation of lung tumors by PET/CT images is respiratory motion. The partial volume effects related to the resolution of the PET/CT scan and the motion can cause inaccuracies for small lesions. To increase measurement accuracy, further studies should consider respiratory movements (using new and more accurate PET/CT devices) and lesion sizes. Nguyen et al. [96] found a need for interpolation at the boundaries of the segmented marrow compartment to account for the physical size difference between voxels in the PET and CT modalities. Clinical practice is limited to a small number of manually delineated ROI in 18F-fluoro-L- deosythymidine SUV measurement. Moreover, thresholding is not the best method for detecting the boundaries of these lesions. Therefore, Perk et al. [70] suggested a statistically optimized regional thresholding (SORT) method for bone lesion detection in 18F-NaF PET/CT imaging. Some patients appeared to have higher healthy bone uptake levels; therefore, the false positive rate in such patients may be elevated.

### 4.3. Clustering/Classification

Classification is a supervised learning technique aimed at partitioning a feature space derived from an image using labels provided for training. Clustering methods group the feature space into regions or proposed classes without labels. These techniques generally do not incorporate spatial information unless it is included in the feature space derivation. Examples of classification and clustering methods used to segment tumors include Random Forest (RF) [2,87,103], Support Vector Machines (SVM) [51,55,63,103,109], fuzzy clustering [101], Decision Tree (DT) [63,67], K-nearest neighbors (KNN) [67], K-means [73], parallelepiped classification [38], and Fuzzy C-Mean (FCM) [69].

Two studies derived useful information regarding the classification process and the ground truth values. Chu et al. [2] developed an RF classifier to segment tumors on bone scans using intensity and context features aimed at addressing areas prone to false positives and found that context features played a critical role. Furthermore, their study performed well in areas where tumors and high-intensity non-tumors were in close proximity, which could be due to the restrictiveness of a rule-based approach compared to a learning-based approach. Markel et al. [67] addressed the challenge of determining ground truth when validating the image segmentation method. They used the simultaneous truth and performance level estimation (STAPLE) algorithm to combine the GTVs into probabilistic maps for each patient. The results showed that all of the algorithms they tested performed better with respect to the test data, as opposed to the training data, which is indicative of a more reliable ground truth. They also showed that the use of texture features within PET/CT images was a promising approach for target delineation in radiotherapy of the lung. Hinzpeter et al. [42] conducted a proof-of-concept study to investigate whether the radiomics of CT image data enables the differentiation of bone metastases using 68 Ga-PSMA PET imaging as a reference standard. The trained gradient-boosted tree achieved an accuracy of 0.9 when applied to its original, non-augmented dataset.

Some authors used hybrid methods, such as combining SVM with either wavelet transform, Naïve Bayes, or DT [55,109]. AbuBaker and Ghadi proposed a novel algorithm for the detection and enhancement of cancerous nodules in CT images using SVM and wavelet transform. The use of both wavelet and SVM features reduced the predicted false positive regions in the processed CT images in their study. Hussain et al. [63] presented an automated lung cancer detection system based on multimodal features, such as texture, morphological, entropy-based, scale-invariant Fourier transform (SIFT), and Ellipse Fourier Descriptors (EFDs), using machine learning techniques, such as SVM, Naive Bayes, and DT. Wiese et al. [51] detected sclerotic bone metastasis in the spine using watershed algorithm and SVM. Complexity due to the heterogeneity, less isolation, and additional lesions in the single clinical case was addressed by training SVM on 3D features and imposing additional constraints (overlap and intensity) during the merge into three dimensions. The proposed model could increase the sensitivity in the initial detection of sclerotic metastatic lesions in the spine and in the assessment of bone tumor burden in cases of known sclerotic bone metastasis.

The drawbacks in these studies can be found throughout the segmentation process. Markel et al. [67] identified that in the preprocessing stage of segmenting lung cancer, a tumor may present a necrotic core with a low uptake, which resulted in small cavities in the segmentation. As a solution, they introduced a fill procedure in the post-processing step, and this worked well in the segmentation, as segmentation is a closed shape. Furthermore, they suggested incorporating 4D-PET images to better coincide with gated CT images to reduce motion blurring. Naqiuddin et al. [69] segmented CT images into bone, brain, and tumor regions using an FCM algorithm. They identified a sample size issue and gender bias in their dataset. Generally, fuzzy clustering techniques exclude spatial information when assigning associations to individual data even though it performs well in categorizing heterogenous data. This can be an issue, as medical images present a high degree of spatial correlation between tissues and the technique is sensitive to noise. This issue was also addressed by Slattery [101] using an additional membership function to include spatial information. In the Wiese et al. [51] model, to detect sclerotic bone metastases, some lesions were missed due to the weaknesses of the watershed algorithm. The feature filter eliminates some true detections due to the intensity contrast of the lesion with the surrounding osseous material. After watershed segmentation, the authors implemented a merging routine as a solution to over-segmentation. Polan et al. [87] observed a limitation in segmenting tissues using RF. The trainable Weka segmentation (TWS) tool limits the optimization of the algorithm. As a result, the number of trees and leaf size of the classifier ensemble were not optimized. Further, the creation of voxel features for training and classification in TWS was limited to the minimum, maximum, mean, and variance of the region of voxels.

### 4.4. Statistical Methods

Various studies have employed statistical methods such as the Active Shape Model (ASM) [48], a Bayesian delineation framework [103], gradient based segmentation [57], geometrical shape model under Bayesian framework [102] and fuzzy Markov random field (MRF) model [62].

Rachmawati et al. [48] utilized ASM to segment the cancer metastasis of bone scan images of Indonesian patient data, which resulted in the shape estimation of each predefined region of the bone scan image. Sheen et al. [123] compared the fixed thresholding-based method with gradient-based edge detection to compare radiomic signatures and prediction models. The results showed that gradient-based edge detection derived significant radiomic features for the model.

Both Ninomiya et al. [103] and Guo et al. [62] used Bayesian models. Ninomiya et al. [103] used an anatomical features-based machine learning technique to develop a Bayesian delineation framework of Clinical Target Volumes (CTV) for prostate cancer. One of the drawbacks of the Bayesian approaches in this scenario is the localization of the CTV to place probabilistic atlases (PAs). This Bayesian framework did not work well when the CTVs were far from the average CTV position. The proposed framework, using anatomical-features-based machine learning (AF-ML), more accurately extracted the CTVs of prostate cancer. Additionally, Guo et al. [62] utilized a fuzzy MRF model to segment lung tumors on PET/CT images. Unlike the traditional fuzzy MRF model method, it utilizes a new joint posterior probabilistic model, which can effectively take advantage of both CT and PET image information for the identification and delineation of tumor volume. For lung tumors located near other tissues with similar intensities in PET and CT images, such as when they extend into the chest wall or the mediastinum, this method was able to achieve more effective tumor segmentation.

Geographical bias was an issue in some of the studies in this review. In Rachmawati et al. [48], training data from non-Indonesian patients improved generalizability. If the bone geometry of the bone scan image has too many variations across countries, it might degrade the accuracy of metastasis detection because bone geometry is strongly influenced by ethnicity [48,124]. In the study by Zhou et al. [109], under the deep learning section, they demonstrated geographical bias when their study was limited to only Asian patients.

### 4.5. Atlas-Based Approaches

Segmentation is often limited by the low contrast between adjacent tissues, but prior knowledge can improve this. A widely used method is to incorporate prior knowledge from a reference image called an atlas, which provides an estimate of an object’s position much like a map would describe the components of a geographical area, and it helps to distinguish adjacent objects of interest with similar features [16].

Some authors utilized multiple atlases [78,125], while others used a combination of different techniques with a specific atlas [84,91]. Hanaoka et al. [83] utilized multiple atlases registered to the target unseen volume by a novel landmark-guided diffeomorphic demons’ algorithm, segmenting the whole spine and pelvis in a CT image. One of the advantages of this algorithm is the diffeomorphism/invertibility of the deformation field. Invertibility is required if it is necessary to warp both image(s) and landmark(s). The deformation field for warping images is not the same as and is the inverted version of the field for warping landmarks. Furthermore, multi-atlas-based approaches used by Yusufaly et al. [97] may allow active bone marrow sparing in radiotherapy settings where PET/CT is unavailable. Although previous experience strongly suggests that active bone marrow sparing is causally related to a reduction in hematologic toxicity, more outcome data are required to conclusively verify the benefits of an atlas-based approach. Fritscher et al. [91] utilized an atlas-based segmentation approach in combination with label fusion to initialize a segmentation pipeline employing statistical appearance models and geodesic active contours. The proposed hybrid approach, Multi Atlas Based Segmentation (MABS), provided more accurate results within a clinically acceptable amount of time, even in the presence of noise and low image contrast. However, MABS lacks the ability to provide anatomically plausible segmentation results and partly also shows less accurate results near boundaries. Another hybrid approach by Ruiz-España et al. [88] was developed for the automatic segmentation of the vertebrae from CT images by combining two different segmentation methods using the level-set method and probabilistic atlas. However, the generalizability of this method to other structures for which clear anatomical feature points are less reliable needs to be investigated.

Several limitations were observed in the atlas-based approaches. The test dataset in the study of Hanaoka et al. [83] only included healthy spines or those with osteoporosis. Spines with scoliosis, lordosis, postsurgical changes, or bone metastasis were not included. Another problem is that spines with abnormal numbers of vertebrae were excluded from the dataset, even though such anatomical anomalies are quite common [78]. A typical problem for geodesic active contours (GAC) is leaking of the evolving contour into neighboring structures in the absence of strong boundaries, which can be avoided by combining GAC with InShape models. The downside of many model-based segmentation approaches is their susceptibility to local minima. This has been overcome using MABS as an initialization restricting the final optimization to a local search space where all quantitative tests were carried out using a leave-one-out cross validation [91]. To improve the accuracy and efficiency of atlas-based auto-segmentation methods, further implementation of and investigations into artificial intelligence with deep learning algorithms are needed. Further investigations into the feasibility of RT plans based on ABAS-generated contours for both CTV and OAR are also needed [64].

### 4.6. Region-Based Approaches

Region-based methods can be categorized into region-growing or graph-based methods, which consider homogeneity when determining the object boundaries [14]. The main assumption in region growing for segmentation is that the region of interest has nearly constant or slowly varying intensity values to satisfy the homogeneity requirement. This method incorporates spatial information along with the intensity, which is an advantage over thresholding. However, different homogeneity criteria and initial seed locations can affect the segmentation results and require tuning [14].

Region-growing methods, similar to thresholding, are sensitive to the noise in the image, and can lead to leakage. The method proposed by Yang et al. [76] addressed this issue by developing a lung tumor segmentation based on multi-scale template matching and region growing; however, the sample size limited the evaluation of the technique. Dong et al. [81] proposed a method combining mathematical morphology based on a labeling algorithm and Graph Cuts to segment vertebrae in 100 sliced images of 10 patients with bone metastasis. The proposed method outperforms the conventional graph cut method. For lung segmentation, Elsayed et al. [61] applied a region-growing technique to isolate the human body and then a threshold followed by the Hessian method for vascular tree segmentation. This precisely extracted nodule features. Several classifiers and their combination were applied to classify malignant or benign nodules.

Meanwhile, Graph Cuts have the advantage of realizing fast and accurate segmentation of the target with little intervention of radiologists, as this method utilizes both boundary and regional information [81].

## 5. Conclusions

This paper analyzed the literature on tumor segmentation approaches, with a focus on bone metastases. We found that the development of segmentation techniques that combine anatomical information and metabolic activities (e.g., PET/CT) shows encouraging results. However, the lack of a gold standard for tumor boundaries is a major hindrance to the acceptance of fully automatic segmentation. Most algorithms need manual correction of the segmentation output, which largely explains the absence of clinical translation studies. AI-based methods may be better suited as an assistant for the clinician to overcome the repetitive and time-consuming task of identifying and segmenting lesions while providing a measurement of whole-body tumor burden.

To fully understand the methods and algorithms that can be utilized to deliver proper treatment planning to individual patients, more comprehensive studies are required that have limited data biases. When developing AI-based methods, it is crucial to utilize an appropriate method as a baseline approach. The nn-Unet framework is a state-of-the-art deep learning method with the capability to automatically set hyper-parameters while considering factors such as input data features and memory consumption. Utilizing simple thresholding as a baseline method prior to utilizing more complex methods is also advisable as it provides an understanding of the images’ features and the ability to conduct independent experiments.

Open-source software, such as a 3D slicer, can be used for initial visualization and segmentation tasks, as it is a reliable platform for medical image analysis, visualization, and clinical support. It is beneficial for the research community to recreate and build upon previous findings in similar research areas with different datasets. Although researchers need to make their code available for this to occur, many of the reviewed papers lack readily available codes. It is recommended that code in future studies be made readily accessible in open-source formats.

## Figures and Tables

**Figure 1 cancers-15-01750-f001:**
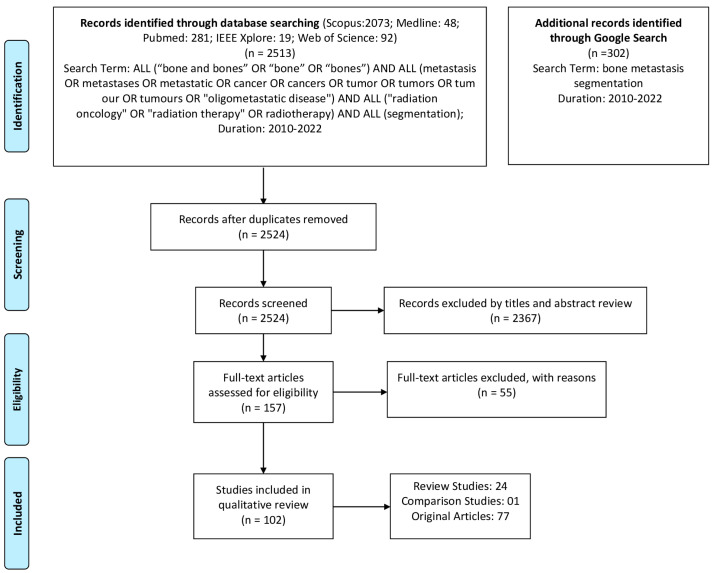
Inclusion and exclusion of articles for the review.

**Figure 2 cancers-15-01750-f002:**
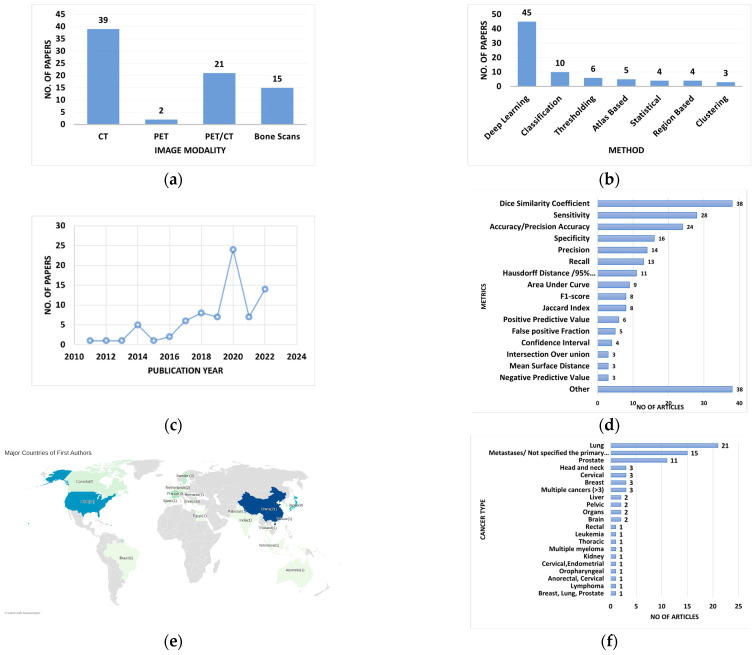
Analysis of characteristics of included articles. (**a**) Distribution of articles according to the image modality; (**b**) Distribution of articles according to the method; (**c**) Distribution of articles over years; (**d**) Distribution of evaluation metrics; (**e**) Countries of first authors; (**f**) Distribution of cancer type.

**Table 1 cancers-15-01750-t001:** Papers Included in the Review.

Area of the Study	Purpose of the Study	Reference	No of Papers
Reviews/Comparison of methods	Computerized PET/CT Image Analysis in the Evaluation of Tumor	[11]	1
	Machine learning techniques in medical imaging	[19,20,22,27,28,29,33]	7
	Segmentation methods for Radiology image (s)	[14,16,18,21,23]	5
	Radiation therapy treatments for metastases	[4,5,6]	3
	Radiation therapy and planning	[9,10,12,34,35]	5
	Metastases Segmentation	[26]	1
	Imaging Techniques	[17,36]	2
	Radiomics in medical imaging	[25]	1
Segmentation	Metastases	[37,38,39,40,41,42,43,44,45,46,47,48,49,50,51,52,53,54]	18
	Tumor	[2,55,56,57,58,59,60,61,62,63,64,65,66,67,68,69,70,71,72,73,74,75,76,77,78,79,80]	27
	Organ(s)/Organs-at-Risk (OARs)	[81,82,83,84,85,86,87,88,89,90,91,92,93,94,95,96,97,98,99,100,101,102]	22
	Target Volume/OARs + Target Volume	[103,104,105,106]	4
Classification	Metastases	[13,107,108,109]	4
	Tumor	[110,111]	2
**Total**			**102**

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
