# Peer review of "Systematic Review of Tumor Segmentation Strategies for Bone Metastases"

_cancers, 2023, doi:10.3390/cancers15061750_

Round 1

Reviewer 1 Report

The main objective of this review is to present an overview of the latest research on cancer segmentation, and bone metastasis segmentation on radiology images in the context of radiation therapy planning and to analyze and compare with the state-of-an techniques in computer versions. This review article will provide useful information in the field of medical science related to bone metastasis. This reviewer recommends that the article is suitable for publication in this journal.

Author Response

Response to Reviewer 1 Comments

Point: The main objective of this review is to present an overview of the latest research on cancer segmentation and bone metastasis segmentation on radiology images in the context of radiation therapy planning and to analyze and compare with the state-of-an techniques in computer versions. This review article will provide useful information in the field of medical science related to bone metastasis. This reviewer recommends that the article is suitable for publication in this journal

 Response: Thank you for your positive feedback and recommendation for publication. We are pleased to hear that our review article will provide useful information in the field of medical science related to bone metastasis segmentation in the context of radiation therapy planning. We hope that this review will contribute to the advancement of research in this area and assist in improving clinical practice. We thank you for your time and effort in reviewing our manuscript.

Reviewer 2 Report

Comments on cancers-2252961

In this study, the author has studied “Systematic review of tumor segmentation strategies for bone metastases.This is an engaging article with a robust approach that purposefully questions our knowledge of the subject. However, the presentation of the methodology is somewhat confusing, and the readability of the results could be improved. Addressing both these issues will make this interesting paper more impactful. The English language used in the manuscript needs some improvements as some punctuation, and grammatical mistakes are present. Experimental designs required more clarity. Moreover, research results are not discussed in an understandable manner, reflecting that the author needs a more comprehensive way of thinking.

Specific comments:

1.      The authors are advised to critically revise the abstract, especially the results section.  

2.      Please add more strong keywords and avoid the words used in the title.

3.      Page 2, line 89-90: “To date, no review has presented the bone metastasis segmentation approaches and tumor segmentation with PET, CT, or PET/CT in radiation therapy to our knowledge.” The authors are sure about this statement? Because several studies published about tumor segmentation with PET/CT.

4.      The results section is incomplete and needs more justification.

5.      Page 7, line 228-231: “Other augmentation techniques such as rescale [13, 32, 38, 42-44, 53, 57-59, 97, 106] , rotation [13, 32, 38, 42-44, 53, 57-59, 76, 97, 106], zoom [13, 43, 106], intensity shifts [57, 76], horizontal reflections [76], image translation [38, 42, 76], cropping [58], elastic deformations [44], gamma augmentation[44], and flip [13, 43, 53, 58, 106] have also been used.” It is not appropriate to use too many references in a single sentence.

6.      Page 8, line 259: “final fully connected deep neural network for integrating global (i.e., whole-body) and local (i.e., patches) information…” Please avoid ‘etc., e.g., i.e.,’ in a scientific manuscript.

7.      The conclusion section is too long. Please revise it and make it up to 250-300 words. It is also suggested to remove the references from this section.

8.      It is advised please shift a Supplementary Table to the manuscript (if possible) to strengthen it and added it before the conclusion for a better understanding of readers. Because this table draws an overall story of the manuscript in a comprehensive way.

9.      Authors are advised to proofread the manuscript to overcome grammatical mistakes.

10.  Authors are advised to revise a few subheadings.

Reviewer 3 Report

In this paper, authors investigate methods for differentiating benign from malignant bone lesions and characterizing malignant bone lesions specifically in the context of bone metastases. The following review comments are recommended, and the authors are invited to explain and modify.

Comment: “However, many studies did not provide evidence that their proposed methods are suitable for use in clinical practice”, what is concrete justification of this after this review work?

Comment: “Literature published from January 2010 to March 2022 were considered. A total of 2524 papers were identified, and 77 were included in the review”, but Figure 1 shows n=102?

Comment: “Figure 2(b) Distribution of articles according to the method”, that should be some supervised/unsupervised techniques? What is justification of “classification” and “(e) Countries of first authors” in Figure?

Comment:

The following clinical decision support systems using AI, and medical imaging must be included to improve the quality of the segmentation strategies of this review paper (10.3390/math10050796).

Comment: There are already many review articles on this topic; how’s their work is different and what enhancement done in it? Like as:

1.        “Bone Metastases: An Overview”.

2.       “Artificial intelligence performance in detecting tumor metastasis from medical radiology imaging: A systematic review and meta-analysis”

3.       “Artificial Intelligence in Bone Metastases: An MRI and CT Imaging Review”

Comment: Organization of paper looks like as a research article but it is review paper.

Round 2

Reviewer 2 Report

Based on the response file, the authors have carefully revised the manuscript but in the main file, the authors did not highlight the changes. In the future, it is suggested to highlight the changes made in the revised version of the manuscript for a better understanding of reviewers. 

Reviewer 3 Report

The authors have answered my questions satisfactorily.